# Evolution of a research field—a micro (RNA) example

Máire-Caitlín Casey, Michael J. Kerin, James A. Brown and Karl J. Sweeney

Discipline of Surgery, School of Medicine, National University of Ireland, Galway, Ireland

## ABSTRACT

**Background.** Every new scientific field can be traced back to a single, seminal publication. Therefore, a bibliometric analysis can yield significant insights into the history and potential future of a research field. This year marks 21 years since that first ground-breaking microRNA (miRNA) publication. Here, we make the case that the miRNA field is mature, utilising bibliometrics.

**Methods.** Utilising the Web of Science™ (WoS) database publication and citation information, we charted the history of miRNA-related publications, describing and dissecting contributions by publication type (plus category, pay-per-view or open access), journal (highlighting dominant journals), by country, citations and languages.

**Results.** We found that the United States of America (USA) publishes the most miRNA papers, followed by China and Germany. Significantly, publications attributed to the USA also receive the most citations per publication, followed by a close grouping of England, Germany and France. We also describe the relevance and acceptance of the miRNA field to different research areas, through its uptake in areas from oncology to plant sciences. Exploring the recent momentous change in publishing, we find that although pay-per view articles vastly out-number open-access articles, the citation rate of pay-per-view articles is currently less than double that of open-access.

**Conclusions.** We believe the trends described here represent the typical evolution of a research field. By analysing publications, citations and distribution patterns, key moments in the evolution of this research area are recognised, indicating the maturation of the miRNA field and providing guidance for future research endeavours.

Corresponding authors
James A. Brown,
james.brown@nuigalway.ie
Karl J. Sweeney,
karl.k.sweeney@nuigalway.ie

## INTRODUCTION

With expanding scientific production and unprecedented access to information, a means of independently assessing and analysing research output becomes apparent and essential. Consequently, various data analysis tools and internet-based search engines have been devised to enable the processing and organisation of this scientific output into a manageable form. Bibliometric parameters are one such tool utilised for this assessment. First described by Paul Otlet in 1934 (*Rousseau, 2014*), bibliometrics refers to the quantitative statistical analysis of publications to enable activity and dynamics within research fields to be mapped.

The onset of the digital age revolutionised the manner in which scientific knowledge is produced and distributed. *Gibbons et al. (1994)* provide an account of this fundamental change in their concept of '*Mode 2*' knowledge production. This refers to the development of a highly interactive and transdisciplinary research system that is socially distributed. *Nowotny, Scott & Gibbons (2001)* elaborated this concept still further, highlighting novel and contemporary scientific practices with an increasing range of '*knowledge producers.*' Various theories of science evolution have been proposed, with some authors analogising the progression of research to the evolution of living organisms: the introduction of new concepts, development of novel research directions and the emergence and loss of hypotheses (*Chavalarias & Cointet, 2013*).

The field of microRNA research presents an exceptional opportunity to observe the progression of a novel area of scientific investigation from point of discovery to rapidly maturing field, using bibliometrics. The term microRNA (miRNA) was coined by *Ruvkun (2001)*, to refer to a naturally-occurring class of short, non-coding RNA molecule between 19 and 21 nucleotides long. *Lee, Feinbaum & Ambros (1993)* discovered the first miRNA in 1993, isolating *Lin-4* from the nematode *Caenorhabditis elegans*. It took seven years before a second miRNA, *Let-7*, was discovered by *Pasquinelli et al. (2000)*. The revelation that *Let-7* sequence, expression and function were conserved across animal phylogeny (*Pasquinelli et al., 2000*), from nematodes to humans, resulted in a research revolution. Subsequently, thousands of miRNAs have been identified in eukaryotes, including plants, fish, fungi and mammals. In humans alone approximately 2,555 unique mature miRNAs have been identified (http://www.mirbase.org/). While the exact function of all recognized miRNAs remains to be fully elucidated, they are known to regulate gene expression via binding target messenger RNA (mRNA), inhibiting translation or triggering mRNA degradation. Importantly, it has been demonstrated that in addition to their inhibitory role miRNAs can function to induce or activate transcript levels (*Vasudevan, Tong & Steitz, 2007*; *Place et al., 2008*). Through these mechanisms miRNAs fulfil a regulatory role in various cellular processes, including cell development, differentiation, proliferation and apoptosis (*Place et al., 2008*; *Filipowicz, Bhattacharyya & Sonenberg, 2008*; *Bartel, 2004*).

Deregulated miRNA expression patterns have been noted across all organisms, encompassing a wide spectrum of pathological processes, from immunological defects in fish, altered developmental phase transition and flowering time in plants, and neurode-generation, cardiovascular disease and cancer in humans (*Calin et al., 2002*; *van Rooij et al., 2006*). Discovery of deregulated miRNA expression led to the hypothesis that miRNAs could potentially be used as diagnostic or prognostic markers of disease. Furthermore, miRNA are attractive therapeutic targets for the treatment of various conditions, including cancer. The novel role of miRNA and their importance to many different processes has led to an explosion into the scientific enquiry of miRNA function. The aim of our study is to utilise one data analysis tool, Web of Science™, in conjunction with current theories of research evolution, to quantitatively analyse a novel field of investigation from its point of discovery, to outline its progression and potentially predict its future course.

## MATERIALS AND METHODS

### Database

Citation data was retrieved from the *Web of Science*[TM] (WoS) database (http://apps.webofknowledge.com/), produced by Thomson Reuters (see results section, "database"). Search results from WoS encompassed entries from the *WoS Core Collection*, comprising *Science Citation Index Expanded* (SCI-EXPANDED), *Social Sciences Citation Index* (SSCI), *Arts & Humanities Citation Index* (A&HCI), *Book Citation Index* (BKCI), *Conference Proceedings Citation Index* (CPCI), *Current Chemical Reactions* (CCR Expanded) and *Index Chemicus* (IC) (http://images.webofknowledge.com/WOKRS513R8.1/help/WOS/hp_database.html).

### Search terms and methods

The WoS database was searched utilising the terms "miRNA" and "microRNA" with the Boolean operator "OR". Upon analysis of initial search findings, conflicting results were identified, namely publications containing the following: *mirna estuary, mirna bay, mirna river, mirna equation, Mirna A (author)* and *mirna SC*. As such, the Boolean operator "NOT" was utilised to exclude these findings and refine our search results. Further to this, the research category of "*Agriculture*" was excluded from our search, as several inaccurate search results were identified within this classification.

Although recognised as the first miRNAs discovered, the initial publications pertaining *Lin-4* and *Let-7* did not utilise the terms microRNA or miRNA (coined in *Ruvkun, 2001*), as such these papers did not feature in the search results. To account for this, the *Web of Science Core Collection* database was searched for *Lin-4* and *Let-7* in isolation, utilising the Boolean Operator "NOT" to exclude miRNA and microRNA, so as to prevent overlap with original search results.

The publication timeframe analysed encompassed January 1993 to December 2013. Publications of all languages were accepted, comprising all peer-reviewed articles, including reviews, letters to the editor and editorials.

### Data analysis

WoS data tools were utilised to perform certain elements of result analysis e.g. generating *Journal Citation Reports*. Additional data analysis was performed using *Microsoft Excel 2010©* and *Minitab* version 16 [®].

N.B. The results returned from WoS upon searching study criteria were found to increase with the passage of time. This is thought to be due to delayed indexing of journals among other factors. As a result, some figures containing quantitative numbers may differ slightly among sections of this manuscript, when totalled.

## RESULTS

### Database

Several research platforms currently exist for examining bibliometrics, including the Web of Science[TM], Highwire© and PubMed[TM]. Prior to commencing this study, the suitability

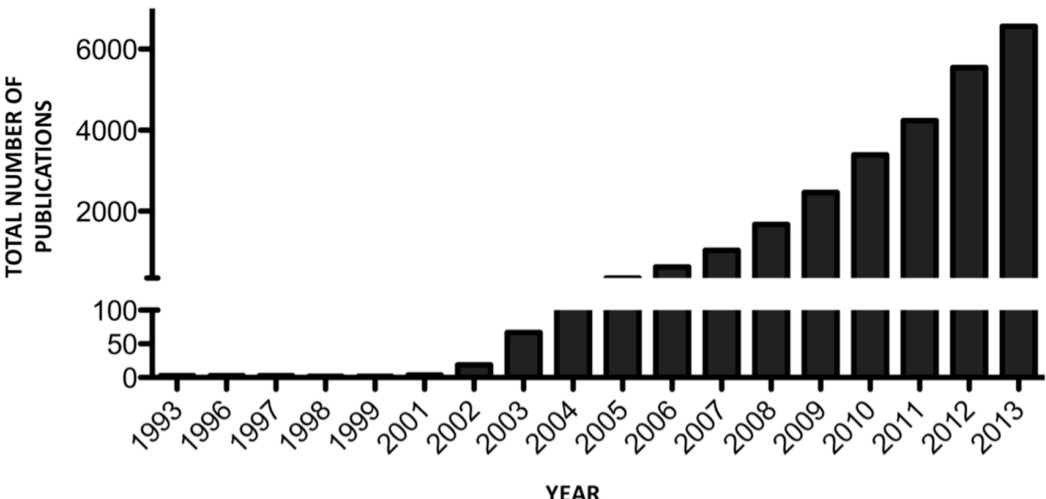

**Figure 1  Number of miRNA publications per year.**

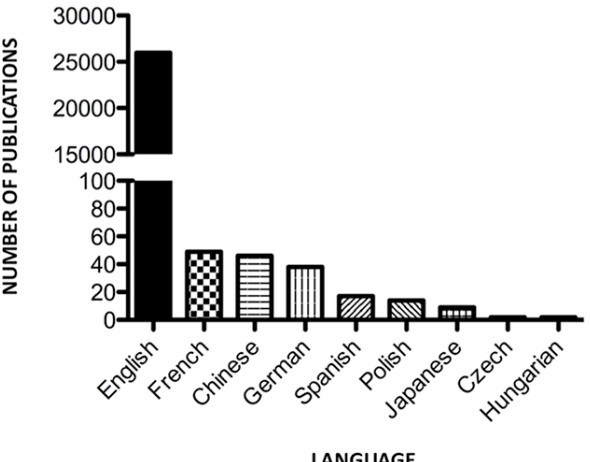

**Figure 2  Language of publication.**

of these databases was individually analysed to allow selection of the optimal data resource. Ultimately, WoS was chosen for the purpose of this study due to superior journal coverage, approximately 9,300 compared to HighWire (1,700) and PubMed (5,669). An additional significant factor was the return of a substantially higher proportion of results using the defined study search criteria (see section 'Search terms and methods').

## Publication distribution

The number of published items identified pertaining to miRNAs, as catalogued in the Web of Science[TM] Core Collection database (1993–2013), totals 26,177 publications. The first publication occurred in 1993, with minimal additional publications (35) until 2003 (Fig. 1). Sixty-two percent of all miRNA literature (16,348 publications) was published within the period 2011–2013 inclusive. Currently, the highest output occurred in 2013,

**Table 1** Rank of 25 countries publishing miRNA material most prolifically with cumulative number of publications per country.

| Rank | Country | No. publications |
| --- | --- | --- |
| 1 | United States | 11,056 |
| 2 | People's Republic of China | 5,584 |
| 3 | Germany | 2,083 |
| 4 | United Kingdom | 1,527 |
| 5 | Japan | 1,474 |
| 6 | Italy | 1,455 |
| 7 | Canada | 904 |
| 8 | France | 848 |
| 9 | Australia | 650 |
| 10 | South Korea | 645 |
| 11 | Netherlands | 625 |
| 12 | Spain | 621 |
| 13 | Switzerland | 492 |
| 14 | Denmark | 420 |
| 15 | Taiwan | 411 |
| 16 | India | 371 |
| 17 | Israel | 348 |
| 18 | Sweden | 335 |
| 19 | Belgium | 292 |
| 20 | Ireland | 254 |
| 21 | Singapore | 239 |
| 22 | Austria | 203 |
| 23 | Brazil | 201 |
| 24 | Poland | 184 |
| 25 | Greece | 158 |

with 25% of total miRNA publications (6,560). Of total publications identified, 99.2% of items were published in English (25,980 publications), with the remainder of articles in French (49 publications), Chinese (46 publications), German (38 publications), Spanish (17 publications), Polish (14 publications), Japanese (9 publications), Czech (2 publications) and Hungarian (2 publications) (Fig. 2).

## Publications by country

To further analyse miRNA-related literature, the distribution of publications by country was determined. In total, 84 countries contributed to the miRNA literature, as outlined in Table 1 (Top 25). The top 5 most prolific countries account for 83% of total publications (21,652). The USA is responsible for 42% of all miRNA literature (11,056 publications) followed by the Peoples' Republic of China 21% (5,584 publications), Germany 8% (2,083 publications), the United Kingdom (1,527 publications) and Japan 6% (1,474 publications). Further nations featuring within the top 10 most prolific countries include Italy 6% (1,455 publications), Canada (904 publications), France (848 publications), Australia (650 publications) and South Korea (645 publications) (Fig. 3). Interestingly,

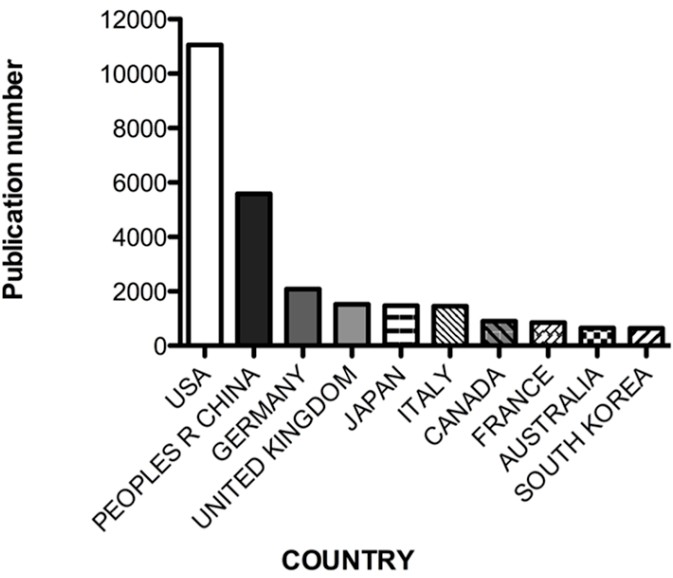

**Figure 3  Number of publications per country.**

separating the United Kingdom into its constituent countries England alone would rank 6th (1,239 publications) behind the USA, the Peoples' Republic of China, Germany, Japan and Italy (Fig. S1).

## Research categories

Analysing the ontological research categories by which miRNA publications are classified in the WoS database, the top 50% of miRNA publications identified can be categorised into medicine and health sciences, which comprises of Biochemistry Molecular biology (23.5%, 6,163 publications), Oncology (15.5%, 4,070 publications), Cell biology (15%, 3,923 publications) and Genetics heredity (10%, 2,554 publications) (Table 2). Further ontological categories of miRNA research listed include horticulture, marine science and entomology.

## Peer-reviewed publications

The categories of research, as defined in Table 2, are comprised of miRNA publications that feature in various international peer-reviewed journals. Table 3 outlines the 25 most prolific journals publishing miRNA research (Table S1, 50 most prolific). In the 20 year period analysed, the youngest journal *Public Library Of Science ONE* (*PLoS ONE*) delivered the largest number of miRNA-related publications at 1,589 (6% of total), with *Nucleic Acids Research* responsible for 489 publications (2% of total) and *Proceedings of the National Academy of Science of the USA (PNAS)* 451 publications (2% of total). We found the top 50 journals were responsible for 38% of total miRNA publications, of which the top 10 journals represent 18% of total miRNA publications identified.

## Document type

MiRNA publications are comprised of various document types including original articles, review articles, news items, editorial material, corrections, reprints, database

**Table 2 Rank of 25 research categories featuring miRNA publications most frequently, with number of publications per research category, and percentage of overall publication.**

| Rank | Category | No. works | % Total works |
|---|---|---|---|
| 1 | Biochemistry Molecular Biology | 6,163 | 23.53 |
| 2 | Oncology | 4,070 | 15.54 |
| 3 | Cell Biology | 3,923 | 14.98 |
| 4 | Genetics Heredity | 2,554 | 9.75 |
| 5 | Science Technology Other Topics | 2,474 | 9.45 |
| 6 | Biotechnology Applied Microbiology | 1,634 | 6.24 |
| 7 | Research Experimental Medicine | 1,623 | 6.19 |
| 8 | Haematology | 1,131 | 4.32 |
| 9 | Cardiovascular System Cardiology | 1,068 | 4.077 |
| 10 | Neurosciences Neurology | 1,057 | 4.04 |
| 11 | Pharmacology Pharmacy | 935 | 3.57 |
| 12 | Gastroenterology Hepatology | 906 | 3.46 |
| 13 | Pathology | 895 | 3.42 |
| 14 | Biophysics | 893 | 3.40 |
| 15 | Plant Sciences | 745 | 2.84 |
| 16 | Immunology | 701 | 2.68 |
| 17 | Developmental Biology | 696 | 2.66 |
| 18 | Chemistry | 575 | 2.19 |
| 19 | Endocrinology Metabolism | 573 | 2.19 |
| 20 | Life Sciences Biomedicine Other Topics | 516 | 1.97 |
| 21 | Virology | 482 | 1.84 |
| 22 | Mathematical Computational Biology | 411 | 1.57 |
| 23 | General Internal Medicine | 344 | 1.31 |
| 24 | Surgery | 333 | 1.27 |
| 25 | Physiology | 308 | 1.18 |

reviews, proceedings papers, letters and meeting abstracts. Of the 26,177 publications identified, 69% of all documents were original research articles (18,111 publications). In addition to original articles, 14% of published items comprised meeting abstracts (3,659 publications), 13% review articles (3,314 publications) and 3% were editorial material (647 publications). The remaining publications comprised of a minimal number of article corrections, news items and reprints. Table S2 shows an analysis of published document types for the top 10 journals which published miRNA-related documents. Constant with the overall trend, articles featured most prominently, comprising 88% of publications for these top 10 journals (4,789 publications), followed by meetings abstracts 8% (393 publications), reviews 1.5% (73 publications), editorial material 0.6% (28 publications) and corrections 0.4% (20 publications). Proceedings papers, letters and database reviews comprise the remainder, providing minimal input.

## Publication citations

For all miRNA related publications, 837,898 citations were found by querying the *WoS Core Collection* Database. In concordance with publication numbers, citations per year

**Table 3 Rank of 25 journals publishing miRNA material most frequently, with cumulative number of publications per journal.**

| Rank | Journal | No. publications |
|------|---------|------------------|
| 1 | PLoS ONE | 1,589 |
| 2 | Nucleic Acids Research | 489 |
| 3 | PNAS | 451 |
| 4 | BLOOD | 432 |
| 5 | Journal of Biological Chemistry | 346 |
| 6 | Biochem Biophys Res Communications | 329 |
| 7 | RNA-a Publication of the RNA Society | 303 |
| 8 | Cancer Research | 302 |
| 9 | BMC Genomics | 296 |
| 10 | Hepatology | 252 |
| 11 | Circulation | 246 |
| 12 | Oncogene | 211 |
| 13 | Faseb Journal | 210 |
| 14 | Cell Cycle | 200 |
| 15 | Febs Letter | 180 |
| 16 | Modern Pathology | 179 |
| 17 | Gastroenterology | 176 |
| 18 | Laboratory Investigation | 171 |
| 19 | Genes Development | 167 |
| 20 | Journal of Virology | 162 |
| 21 | Cell | 159 |
| 22 | RNA Biology | 146 |
| 23 | International Journal of Cancer | 136 |
| 24 | Nature | 136 |
| 25 | Oncology Reports | 136 |

increased exponentially peaking in 2008, at which point citation rate decreased (Fig. 4A). Ranking the total number of citations by country, publications originating from the USA received the highest total number of citations ($n = 475,300$) followed by China ($n = 72,265$), Germany ($n = 71,051$), Italy ($n = 47,084$) and England ($n = 42,970$) (Fig. 4B). Analysing the average citation per item for the top 10 countries publishing miRNA material, the USA retained the first position (44.3 citations per item). However, the remaining positions changed with the second position now held by England (35.9 citations per item) followed by Germany (35.4 citations per item), France (34.4 citations per item) and Italy (34.2 citations per item). Interestingly, of the top ten countries China now displays the least number of citations per item ($n = 14.5$) (Fig. 4C). Examining the citations per item from the top 20 countries (as described in Table 1), the rankings change considerably with Switzerland (13th) now displaying the highest number of citations per item at 55.1, followed by USA (44.3 citations per item), then the Netherlands (39.6 citations per item), England (35.9 citations per item) and Sweden (35.8 citations per item) (Table S3).
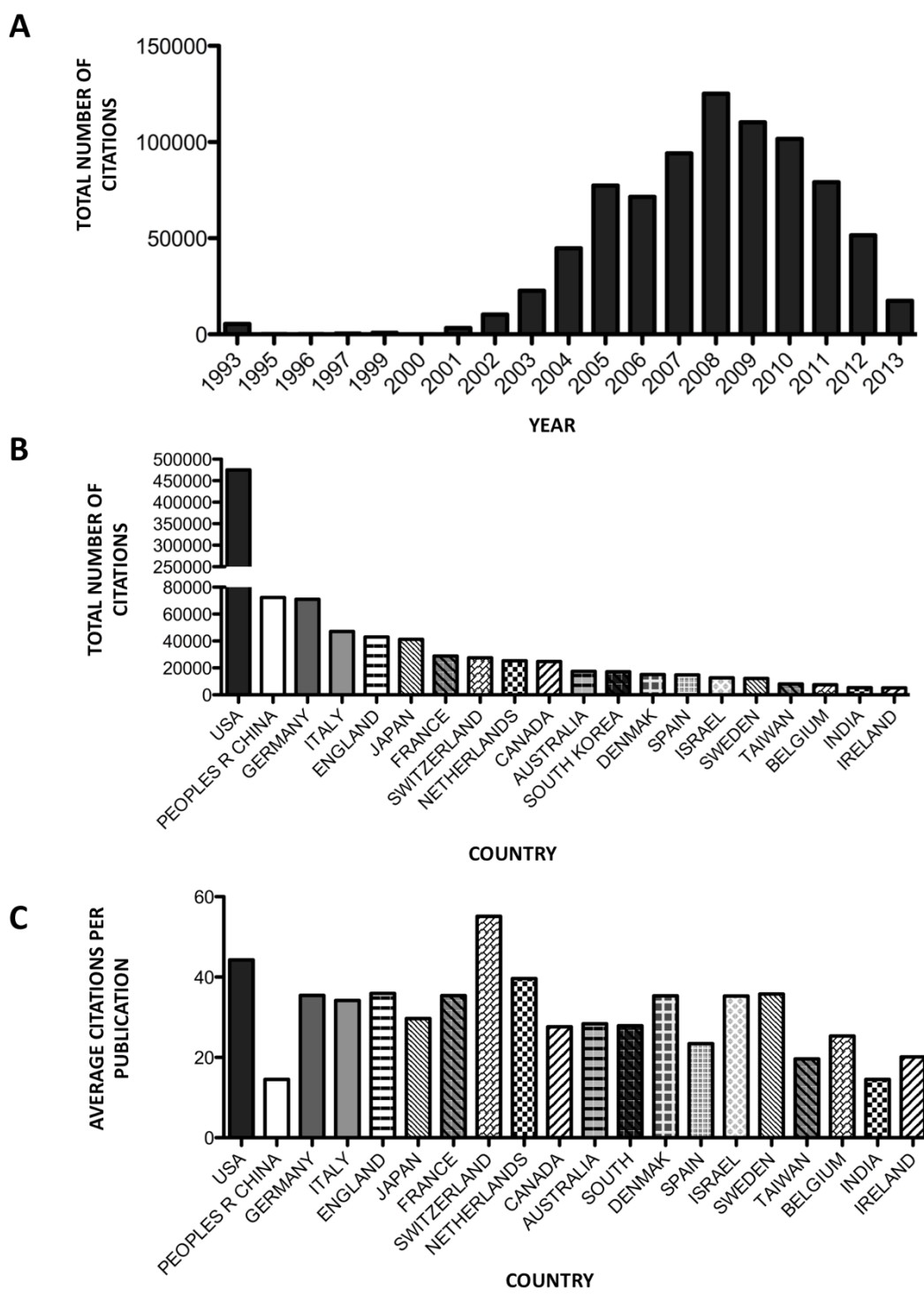

**Figure 4 Publication citations.** (A) Number of citations per yearly publication. (B) Number of cumulative citations per country. (C) Average citation per publication per country.

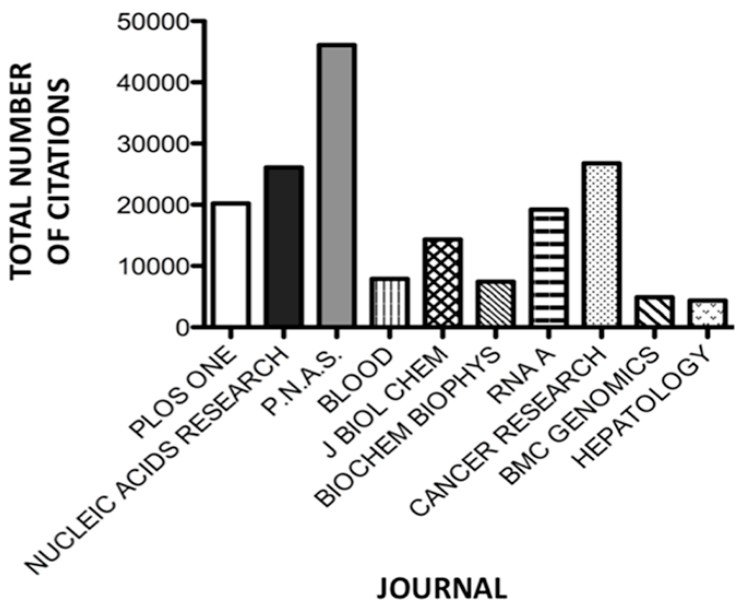

**Figure 5 Number of citations per journal for the 10 journals publishing miRNA material most prolifically.**

To further analyse the citation pattern, we investigated citations of the top 10 journals publishing miRNA material (Fig. 5). *Proceedings of the National Academy of Science of the USA (PNAS)* was the most frequently cited with 6% of the total citations (46,112 citations), followed by *Cancer Research, Nucleic Acids Research* and *PLoS ONE* each with 3% of total citations (26 763, 26 118 and 20 242 citations respectively). The final entry in this list is *RNA A Publication of the RNA Society* with 2% of the total citations (19, 217 citations).

Considering citations per publication, Table S4 outlines the 10 most cited miRNA publications since the discovery of this research field. The top 3 publications cited featured in the journal *Cell*, with an accumulative total of 11,581 citations (1.4% total citations). 6 of the top 10 publications cited featured in *Nature, Nature Reviews Cancer or Nature Methods*, with one publications featuring in *PNAS*. The first, seminal, miRNA publication by Lee et al. ranks third (3,671 citations) and the second key miRNA publication by *Pasquinelli et al. (2000)* features 28th (931 citations). The top 10 cited papers feature over a 16 year period, with three publications in 2005, three in 2006 and one publication in each of the years 1993, 2004, 2008 and 2009. Of this top 10, four items are review articles. To further examine significant miRNA publications, the top 10 most cited primary research miRNA publications were identified (Table 4). *Cell* was still seen to contribute 3 of the top 10, with *Nature, Nature Genetics* and *Nature Methods* contributing 5 publications and *PNAS* one publication, with the addition of one publication by *Science*.

### Open access versus pay-per-view

Of the miRNA related publications identified, 17% of publications were open access ($n = 4,560$), with 83% of publications pay-per-view access ($n = 22,788$) (Fig. 6A). Analysing the citations of these two categories of publication, open access items were cited

**Table 4  Top 10 cited primary research miRNA publications.**

| Rank | Citations | Title | Author | Journal | Year |
|---|---|---|---|---|---|
| 1 | 4,167 | Conserved seed pairing, often flanked by adenosines, indicates that thousands of human genes are microRNA targets | Lewis BP, et al. | Cell | 2005 |
| 2 | 3,671 | The C-Elegans heterochronic gene Lin-4 encodes small RNAs with antisense complementarity to Lin-14 | Lee RC, et al. | Cell | 1993 |
| 3 | 3,512 | MicroRNA expression profiles classify human cancers | Lu J, et al. | Nature | 2005 |
| 4 | 2,337 | Mapping and quantifying mammalian transcriptomes by RNA-Seq | Mortazavi A, et al. | Nature Methods | 2008 |
| 5 | 2,291 | A microRNA expression signature of human solid tumors defines cancer gene targets | Volinia S, et al. | PNAS | 2006 |
| 6 | 2,168 | Microarray analysis shows that some microRNAs downregulate large numbers of target mRNAs | Lim LP, et al. | Nature | 2005 |
| 7 | 2,151 | Prediction of mammalian microRNA targets | Lewis BP et al. | Cell | 2003 |
| 8 | 2,009 | Identification of novel genes coding for small expressed RNAs | Lagos-Quintana M et al. | Science | 2001 |
| 9 | 1,954 | Combinatorial microRNA target predictions | Krek A et al. | Nature Genetics | 2005 |
| 10 | 1,851 | The nuclear RNase III drosha initiates microRNA processing | Lee Y et al. | Nature | 2003 |

84,864 times, representing 10% of overall citations. However, pay-per-view publications were cited 731,470 times, which represented 90% of overall citations (Fig. 6B). Average citation per publication reveals 32.1 citations per pay-per-view publication, compared to 18.6 citations per item for open access publications (Fig. 6C).

## Hallmarks of miRNA research

Utilising the bibliometric data retrieved, we identified the key discoveries in the field of miRNA research. The seminal miRNA publication, outlining the discovery of these short RNA molecules, by *Lee, Feinbaum & Ambros (1993)* is certainly the first hallmark of miRNA research. Subsequent to this, recognition of conservation of miRNA sequence expression across animal phylogeny from nematodes to humans by *Pasquinelli et al. (2000)*, with the identification of further miRNAs, also represents a significant key advancement in this field. As previously discussed (section 'Publication citations'), these papers feature 3rd and 28th respectively in the most cited miRNA publications, highlighting their visibility and influence.

Following these crucial findings, discovery of the regulatory roles of miRNAs in various cellular processes, from differentiation to apoptosis, should be considered highly significant in furthering our understanding of the functionality of these short RNA molecules (*Place et al., 2008*; *Filipowicz, Bhattacharyya & Sonenberg, 2008*; *Bartel, 2004*). The next, related key event in the miRNA field was the discovery of deregulation of miRNA expression associated with human diseases, such as cancer (*Calin et al., 2002*; *van Rooij*

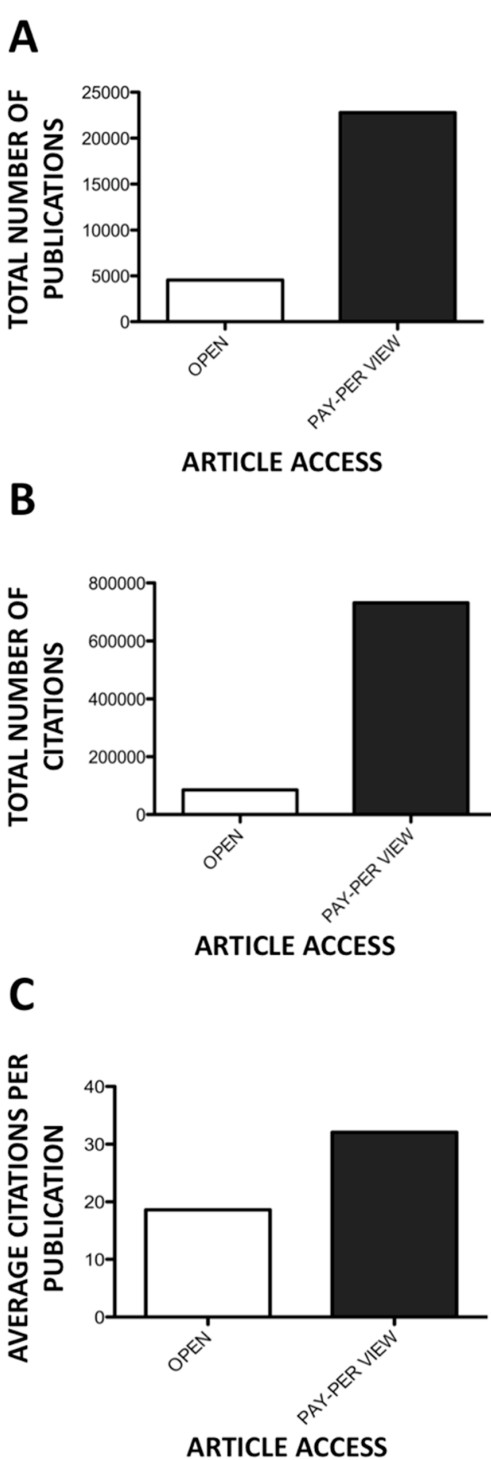

**Figure 6 Open access versus pay-per-view publications.** (A) Number of open access versus pay-per-view publications. (B) Number of cumulative citations by open versus non-open access publications. (C) Average number of citations per open and pay-per-view publication.

*et al., 2006*; *Schaefer et al., 2007*). This discovery raises a potential use of miRNA, as both predictive and prognostic biomarkers of disease. At present, multiple clinical trials are currently registered with ClinicalTrials.gov, investigating the ability of miRNA to function as biomarkers of disease and response to current therapies.

Another key event in the evolution of the field has been the discovery that miRNA are capable of extra cellular signalling (*Valadi et al., 2007*). This novel finding significantly added to our knowledge of the mechanisms that can be employed for cell–cell signalling and communication.

The most recent development in the field that should be considered significant is the therapeutic use of miRNAs as targeted therapies to modulate disease (*Kota et al., 2009*). The attainment of personalised disease management via the use of miRNAs is highly appealing, though many obstacles currently remain, including identification of optimal delivery methods, off-target effects and safety.

## DISCUSSION

To our knowledge, this work represents the first bibliometric analysis of the miRNA field. Our analysis revealed an exponential increase in research output, with yearly publications more than quadrupling between 2005 ($n = 356$) and 2008 ($n = 1,672$), and increasing eighteen-fold by 2013 ($n = 6,560$) (Fig. 1). In describing phylomemetic patterns of science evolution, *Chavalarias & Cointet (2013)* outline the importance of '*special events*' in the progression of a research field, with scientific output increasing significantly during the subsequent time period (*Chavalarias & Cointet, 2013*). *Special events* pertaining to the miRNA field can be identified by analysing the most cited of all primary research miRNA publications since the discovery of this research field (Table 4). The timeline covered by the 10 most cited primary research miRNA publications spans from 1993 to 2008, highlighting the continual key discoveries being made in the miRNA field. Of these 10 publications, 3 were published in the journal *Cell*, with 5 published in *Nature* or one of its subsidiary journals, one publication in *PNAS* and one publication in *Science*. The average *journal impact factor* of the 10 most cited primary research miRNA publications is 27.8, highlighting their visibility and influence as a driving force behind the exponential increase in yearly miRNA publications. Currently, one key event in the evolution of the field was identified that was not found by our bibliometric analysis: miRNA involvement in extracellular signalling (*Valadi et al., 2007*). This publication did not make it into our current top 10 (although it has >1,200 citations), possibly due to its relatively recent publication date (2008). However we anticipate in the future that this publication will enter the top 10 most cited papers in the field. The trend of miRNA publications therefore, adheres to Chavalarias and Cointet's association, with output increasing exponentially in concordance with hallmarks of their brief history—discovery (*Lee, Feinbaum & Ambros, 1993*), recognition of deregulation in cancer (*Calin et al., 2002*), cardiovascular disease (*van Rooij et al., 2006*), autoimmune and neurodegenerative disease (*Schaefer et al., 2007*; *Sonkoly et al., 2007*), potential use as disease biomarker (*Cortez & Calin, 2009*), expression in serum/plasma (*Cortez & Calin, 2009*) and use of anti-miR's as targeted therapy (*Kota et al., 2009*).

While 84 countries have contributed to miRNA literature to date, five countries dominate scientific production in this research field (Table 1). The USA, China, Germany, the United Kingdom and Japan are collectively responsible for more than 80% of all current miRNA literature with 96% of total citations. The distribution of citations per country differs considerably when considering average citations per publication. Switzerland exhibits the highest number of citations per item, with the USA featuring 2nd, followed by a grouping of the Netherlands, England, Sweden and Germany (Fig. 4C). While China features second in the top 10 countries publishing miRNA material, it exhibits the lowest number of citations per publication ($n = 14.5$), reflecting a large portion of uncited literature. Further analysing citations by journal access type (open access journals to pay-per-view), it was observed that pay-wall restricted journals, which represented the majority of publications (83%), accounted for 90% of total citations. Interestingly, the average citation per publication was 1.7 times higher for pay-wall restricted journals compared to open access journals (Fig. 6C). While this adheres to currently observed citation patterns in the literature, due to the growing popularity of open access journals and open-access publication requirements from funding agencies, it is proposed that this discrepancy will no longer be apparent in future years, with open-access publications and citation numbers currently increasing (*Bjork & Solomon, 2012*).

Of the miRNA publications identified in the WoS database, it is interesting to note that 69% of all documents were original articles, reflecting the relative youth of this novel field of investigation, with only ∼10 years of sustained multi-group research efforts. In an analysis of the progression of a field of science, *Bonaccorsi (2008)* outlines increasing diversity within research paradigms as an instrumental factor, attributing this to various scientific hypotheses and the investigative techniques applied to examine them (*Bonaccorsi, 2008*). With the discovery of a definitive role for miRNA in the pathogenesis of multiple disease processes, the predominance of medicine and health sciences becomes apparent (Table 2). Categories of published research span the gamut of medical domains, from immunology to oncology, haematology to virology and neuroscience to surgery. While further areas of investigation feature such as entomology and agriculture, their contribution to overall research output is currently negligible.

In an analysis of the dynamic interest in research topics within the biomedical scientific community, *Michon & Tummers (2009)* identified trends that are exemplified by our analysis of miRNAs. When novel research is initially published, it generally appears in high impact journals, followed by a lag in scientific output, prior to subsequent progression of publications. The initial miRNA publication outlining the discovery of lin-4 appeared in 1993 in the journal '*Cell*' which then brandished an impact factor of 37.2. Subsequent to this however, miRNA output did not significantly progress further until 2003. While miRNA publications began to escalate, the average journal impact factor of the top 10 publishing journals was 13.6. Five years later, with miRNA output increasing still further, the equivalent impact factor decreased to 9, with a further drop to 7.4 by 2012. With increasing scientific output, a shift towards lower impact journals is seen, producing a *long-tail distribution* of publishing when viewed by host journal impact factor. Originally

described by Vilfredo Pareto, a social economist, the *long-tail distribution* can refer to a number of observable phenomena. In this context, it is used to describe a publishing pattern whereby high and medium impact factor journals feature in the minority, with the majority of journals having minimal citation impact (*Michon & Tummers, 2009*). Presence of this distribution is recognised as a sign of acceptance of a research topic as valid within the scientific community (*Michon & Tummers, 2009*). Reaching this stage of publication saturation, *Pfeiffer & Hoffmann (2007)* advocate the development of novel research directions within a given field as particularly advantageous, with pioneering work potentiating publication in high impact journals, and thus returning the cycle to the beginning of the *long-tail distribution* once more.

## CONCLUSION

When we consider the ongoing remodelling of scientific production, our analysis of publication trends, citations and distribution patterns was very informative. Recognising the developmental stage of a particular research field provides researchers with direction and guidance, both in current and future investigative goals. The current unprecedented access to scientific material and bibliometric information provides an opportunity to analyse the dynamics of scientific landscapes, enabling the production of informed, targeted scientific outputs.

**Abbreviations**

| | |
|---|---|
| **miRNA** | microRNA |
| **WoS** | Web of Science |
| **PLoS ONE** | Public Library Of Science |
| **PNAS** | Proceedings of the National Academy of Science of the USA |

### Funding

Maire-Caitlin Casey, James A. Brown and Michael J. Kerin are funded by BREST-PREDICT and the National Breast Cancer Research Institute (NBCRI). The funders had no role in study design, data collection and analysis, decision to publish, or preparation of the manuscript.

### Grant Disclosures

The following grant information was disclosed by the authors:
BREST-PREDICT.
National Breast Cancer Research Institute.

### Competing Interests

The authors declare there are no competing interests.

## Author Contributions

- Máire-Caitlín Casey conceived and designed the experiments, performed the experiments, analyzed the data, wrote the paper, prepared figures and/or tables, reviewed drafts of the paper.
- Michael J. Kerin analyzed the data, reviewed drafts of the paper.
- James A. Brown conceived and designed the experiments, analyzed the data, contributed reagents/materials/analysis tools, wrote the paper, prepared figures and/or tables, reviewed drafts of the paper.
- Karl J. Sweeney conceived and designed the experiments, analyzed the data, wrote the paper, reviewed drafts of the paper.

## Supplemental Information

Supplemental information for this article can be found online at http://dx.doi.org/10.7717/peerj.829#supplemental-information.

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
