# Peer review of "Evolution of a research field—a micro (RNA) example"

_PeerJ, doi:10.7717/peerj.829_

## Round 0.1 · original submission · Minor Revisions

Please address the minor revisions suggested by the reviewers.

·

Basic reporting

Fine

Experimental design

Fine

Validity of the findings

No comments

Additional comments

The review entitled “Evolution of a research field- a micro (RNA) example” tracks the journey of miRNA field since its discovery in 1993. It is a well timed and an interesting article that might be read with interest by people working in the miRNA field. Few comments-

1. Line 258- The hallmarks of miRNA research must be discussed as a separate section. It is important.
2. Line 59- ....Scientific knowledge is produced and distributed.
3. Line 79- Authors need to mention that based on current literature miRNAs are known to increase transcript levels as well and give suitable references for the same.

Reviewer 2 ·

Basic reporting

No comments

Experimental design

No comments

Validity of the findings

No comments

Additional comments

When analyzing publications on a per country basis (eg Fig. 1) it is unclear why data from the constituent countries of the United Kingdom (England, Scotland, Wales and Northern Ireland) are considered separately, especially since government Research Council spending occurs on a UK -wide basis. The authors should comment on their reasons for this classification.

---

## Round 0.2 · accepted · Accept

I would like to thank you for addressing the reviewers comments which helped in improving the manuscript.